# Continuum Robots and Magnetic Soft Robots: From Models to Interdisciplinary Challenges for Medical Applications

**DOI:** 10.3390/mi15030313

**Published:** 2024-02-24

**Authors:** Honghong Wang, Yi Mao, Jingli Du

**Affiliations:** 1School of Mechano-Electronic Engineering, Xidian University, Xi’an 710071, China; 2School of Chemical and Material Engineering, Jiangnan University, Wuxi 214122, China; maoyi@stu.jiangnan.edu.cn

**Keywords:** soft robots, magnetic soft robots, continuum robots, interdisciplinary challenges, medical robots

## Abstract

This article explores the challenges of continuum and magnetic soft robotics for medical applications, extending from model development to an interdisciplinary perspective. First, we established a unified model framework based on algebra and geometry. The research progress and challenges in principle models, data-driven, and hybrid modeling were then analyzed in depth. Simultaneously, a numerical analysis framework for the principle model was constructed. Furthermore, we expanded the model framework to encompass interdisciplinary research and conducted a comprehensive analysis, including an in-depth case study. Current challenges and the need to address meta-problems were identified through discussion. Overall, this review provides a novel perspective on understanding the challenges and complexities of continuum and magnetic soft robotics in medical applications, paving the way for interdisciplinary researchers to assimilate knowledge in this domain rapidly.

## 1. Introduction

In ancient times, the carriage was mainly dedicated to the nobles, and the wheels were manufactured from rigid materials. They lacked comfort and were expected to be available for ordinary families. However, after the invention of flexible rubber materials and internal combustion engines, new transportation, such as cars and bicycles, quickly entered the homes of ordinary people. Similarly, although most rigid robots are currently limited to factory applications, these rigid robots have large structures and potential safety hazards. With the development of new materials and driving technology, soft robots applications are like changes in traditional transportation [1,2,3,4,5]. In recent years, the study of soft robotics has garnered widespread attention, primarily focusing on applications in medical fields [6,7], underwater robotics [8,9,10,11,12], manipulation and grasping [13,14], space exploration [15], and operations in confined spaces [16,17,18,19]. Given the diverse range of soft robots, this paper primarily concentrates on applying continuum robots (the robot structure has a flexible continuum backbone (Figure 1(1-0a)) or an equivalent continuum backbone) and magnetic soft robots (robots embedding magnetic media in soft materials (Figure 1(1-0b))) in medical settings (Figure 1(1-1)).

Since the concept of continuum robots was first proposed in the Amadeus deep-sea research project [20,21], significant progress has been made in this field [22,23,24,25]. This paper focuses on tendon-driven (Figure 1(1-2a)) [26,27], multi-rod-driven (Figure 1(1-2b)) [28,29,30], and concentric tube actuation (Figure 1(1-2c)) [31,32,33] applied in medical continuum configurations, as well as composite continuum configurations [34,35,36,37,38] or magnetic soft robots (Figure 1(1-2d,e)) formed by these basic components. In the medical field, continuum robots, due to their compliant configurations, have attracted widespread attention in endoscopic and catheter-based interventional surgeries. Researchers from different disciplines have proposed various solutions based on their expertise. From a technical perspective, this includes structure [39] and manufacturing [40], modeling [41,42], sensing [43,44,45], trajectory tracking [46,47], control strategies [48,49,50], state estimation [51], stability analysis [52], and applications [53,54]. From the viewpoint of the discipline, this encompasses mechanical engineering [55], computer science [56], materials [57,58,59,60], chemistry [61], biology [62], and medicine [63]. Although the gap between academia and applied fields is constantly widening, extensive research across interdisciplinary has laid a solid foundation for the rapid application of continuum robots.

Magnetic soft robots [64,65,66], as an emerging subfield within the science of soft robots, have garnered attention for their remarkable controllability and flexibility of movement driven by magnetic fields [67,68]. It is particularly suitable for microcatheter interventional treatments (Figure 1(1-3a–e)) [69,70] or those constrained by extreme environments [71]. In the medical field, these robots have revolutionized the sector with their exquisite control capabilities, enabling in situ monitoring [72], precise drug delivery [73,74], and targeted navigation [75], thereby significantly enhancing the accuracy and effectiveness of treatments [76]. However, their applications extend far beyond this. Owing to their structural programmability [77,78,79], magnetic soft robots also exhibit vast potential in fields like logistics automation [80] and environmental monitoring [81,82]. With designs that prevent the need for complex electrical connections and the ability to operate in tight or hard-to-reach spaces, these robots offer a unique and effective solution for specific, challenging application scenarios.

Continuum and magnetic soft robots, although both categorized within the realm of soft robots, display unique differences and complementary features in their design philosophies, application domains, and technical realizations. From a design standpoint, continuum robots emphasize structural continuity and flexibility, adapting to various complex and constrained environments [39]. In contrast, magnetic soft robots rely on magnetic fields for control, particularly suited for remote or contactless operation scenarios [83]. In the application sphere, continuum robots, due to their exceptional flexibility and adaptability, find widespread use in medical, disaster relief, and deep-sea exploration fields. Magnetic soft robots, conversely, excel in precise control aspects like catheter intervention [84] and targeted drug delivery [85]. Technologically, continuum robots primarily depend on intricate mechanical structures and power systems, such as tendon or rod actuation, posing significant manufacturing challenges at sub-millimeter scales. On the other hand, magnetic soft robots function through external magnetic fields and magnetic materials, offering solutions that can reach sub-millimeter and even micro to nano levels [25]. Despite their distinct differences, both share commonalities and potential for cross-application, including pursuing higher degrees of freedom, more complex motion patterns, and shared challenges in sensing and control algorithms.

Continuum and magnetic soft robots represent two significant branches within medical robotics, each distinguished by their unique actuation methods and potential applications. Despite the extensive literature available for each type of robot within their respective research domains, there is a notable absence of a comprehensive review that compares and synthesizes them within a unified framework. This paper addresses this gap by exploring the interrelationship and potential complementarity between continuum and magnetic soft robots from a modeling perspective. We aim to facilitate interdisciplinary research methodologies and pioneer new avenues of study through a comprehensive analysis of these two robotic systems. We hope this integrated analysis will provide fresh insights and inspirations for technological innovation and practical applications in medical robotics.

This review mainly explores the interdisciplinary applications of continuum and magnetic soft robots from the perspective of models. In this article’s second and third parts, we focus on technical analysis and build a unified theoretical framework for continuum and magnetic soft robot models layer by layer from the perspectives of topology and group theory (i.e., algebra and geometry). This involves not only the models themselves, but also their strong connection to multiple disciplines. The fourth part turns to interdisciplinary analysis, exploring the critical role of models in interdisciplinary intersections, showing the complexity and importance of solving interdisciplinary problems, and how these models can be extended from specific problems to broader subject areas. The fifth part uses the case analysis method to deeply examine the strategies and methods of Professor Zhao’s team in multi-disciplinary comprehensive research, emphasizing the core value of inter-discipline in promoting scientific and technological progress and expanding application fields. Finally, in the discussion and conclusion sections, we will summarize and reflect on the importance and future potential of continuum and magnetic soft robotics in interdisciplinary environments to comprehensively present our research results and perspectives.

**Figure 1 micromachines-15-00313-f001:**
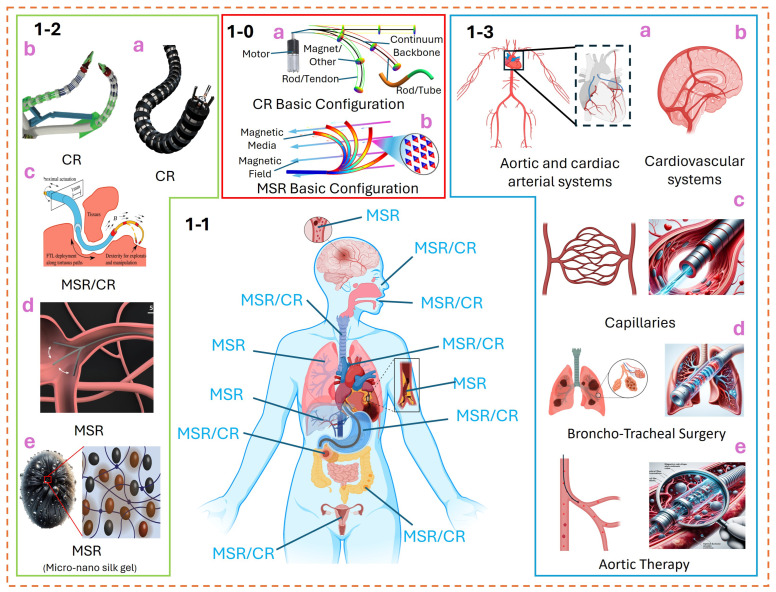
Continuum robots (CR) and magnetic soft robots (MSR) for human medical applications. (**1-0**) The basic configuration of continuum and magnetic soft robots is to initially understand the principles of motion; (**a**) the introduction is the basic configuration of the motion deformation of the continuum robot; (**b**) the introduction is the basic configuration of the motion deformation of the magnetic soft robot. (**1-1**) The sites of action of continuum and magnetic soft robots for applications in human surgery. (**1-2**) The innovative applications of these robotic technologies in medicine, heralding new possibilities in treatment and diagnosis (comprising (**a**) [86], (**b**) [87], (**c**) [35], (**d**) [25], and (**e**) [88], which are reprinted images), further concentrating on several prominent robotic models in the medical sector. The structural type of these robots is the focus of our discussion. (**1-3**) The application of continuum and magnetic soft robots in major human organ surgeries (**a**) in cardiovascular disease surgery; (**b**) in cerebrovascular disease; (**c**) in capillary disease; (**d**) in pulmonary and tracheal disease; (**e**) in aortic and venous vascular disease.

## 2. Continuum Robots

We elucidate the modeling methodologies of continuum and magnetic soft robots through illustrative diagrams and mathematical expressions. This includes exploring principles, data, and hybrid modeling techniques and simplifying the complexity of interdisciplinary integration.

### 2.1. Principle Modeling

Modeling continuum robots is a multifaceted and multi-dimensional challenge. From the perspective of handling the unit structural form, continuum robot modeling can be primarily categorized into several approaches: Cosserat rod theory [89,90,91,92] for micropolar bodies, piecewise constant curvature (PCC) models [23], arc segment models [93], geometrically finite element methods [94], and modal methods [95,96]. Micropolar and finite element approaches are more suited for describing complex nonlinear deformations in continuum robots. At the same time, PCC and arc segment models are better tailored for rapid calculation and control in engineering applications of continuum robots.

Although the Cosserat rod approach, PCC, arc segment models, and modal methods differ in their names and forms of representation, they essentially serve as distinct simplification methods for addressing the same problem. Viewed from the perspectives of group theory and topology [97,98,99], these methods all aim to describe the position and orientation of continuum robots at specific points. Consequently, the kinematic description of continuum robots is fundamentally consistent with that of rigid robots. The particular expressions are as follows:(1)$$C=\left\{g:X\in \left[0,1\right]\mapsto g\left(X\right)\in SE\left(3\right)\right\}$$

In the context of continuum robot modeling, g∈SE(3) encompasses both the position p(X,t) and orientation R(X,t). Precisely depicting the robot’s orientation, including its position and direction, is undoubtedly a central aspect of modeling. Various orientation representation methods, such as rotation matrices, Euler angles, unit quaternions, screw theory [100], and Plücker coordinates, each possess their distinct advantages, limitations, and applicability [101]. The actual choice depends on multiple factors, including the complexity of the application environment and available computational resources. These representation methods can be interconverted through mathematical transformations in certain intricate application scenarios, offering enhanced flexibility. A common method of orientation conversion is presented below:(2)$$R=\exp\left(\theta \hat{K}\right)=I+\sin\left(\theta \right)\hat{K}+\left(1-\cos\left(\theta \right)\right){\left(\hat{K}\right)}^{2}$$

Although rotation matrices are excellent for their intuitiveness, they can be computationally and storage-intensive, which may become a limiting factor in applications of continuum robots requiring real-time control and dynamic simulation. In contrast, Euler angles are easy to understand and implement, but can introduce unnecessary restrictions and complexities in describing complex orientation changes due to the gimbal lock issue. Unit quaternions and screw theory [102,103], within the mathematical framework of Lie groups and Lie algebras, offer more precise and efficient methods for describing the complex motions and configurations of continuum and magnetic soft robots. Lie groups and Lie algebras facilitate a lossless mapping from nonlinear to linear, providing profound and refined mathematical insights into this problem.

From an interdisciplinary perspective, selecting an appropriate method for orientation representation involves a decision-making process that spans multiple dimensions and levels. This decision affects the accuracy and complexity of the model and significantly influences the design of subsequent control algorithms and the optimization of the overall system. Therefore, when making this decision, it is imperative to consider various technical and application factors comprehensively. This interdisciplinary and multi-faceted approach not only aids in advancing fundamental research in continuum robots, but also provides solid theoretical support for their application in various practical scenarios.

In the discussion above, we have detailed the rigid description of robot kinematics. However, given the significant compliance and adaptability of continuum robots, constructing their nonlinear dynamic equations necessitates particular attention to accurately handling the constitutive relations of compliance. In this context, Poincaré’s new dynamics equations provide a critical theoretical framework [104]. Following the criterion of continuity for partial derivatives, $\partial_{t}\partial_{X}=\partial_{X}\partial_{t}$, we can derive the compatibility equations for continuum robots:(3)$$\partial_{X}\eta =-{\mathrm{ad}}_{\xi }\eta +\partial_{t}\xi$$

We have ${\mathrm{ad}}_{\xi }\eta =\left[\xi ,\eta \right]=\xi \eta -\eta \xi$. Observing equations from a temporal or spatial perspective reveals that the velocity field variable η can be expressed as the strain field variable ξ, independent of time *t*. Building upon Equation (Equation 3), it is essential to establish the relationship between strain ξ and the generalized coordinates q. Solid mechanics [105] provides the theoretical underpinning for this relationship. The relationship of the generalized coordinates q can be represented as follows:(4)q=Φ(X)ξ
where Φ(X) is the basis function. To capture the dynamic behavior of continuum robots in complex environments and under the influence of various forces, the kinematic model of continuum robots can be described using the Euler–Lagrange equation or Hamiltonian equation, based on the generalized coordinates q. This kinematic model can be represented as:(5)$$M\left(q\right)\ddot{q}+C\left(q,\dot{q}\right)\dot{q}+G\left(q\right)=\tau_{R}$$

In this context, M(q) represents the mass matrix, $C(q,\dot{q})$ denotes the Coriolis term, G(q) signifies the gravitational term, and $\tau_{R}$ is the input torque. Equation (Equation 5) establishes a more general dynamic equation for continuum robots. To delve deeper into the analysis and synthesis of continuum robots, it is imperative to transform their dynamic model Equation (Equation 5) into a first-order Hamiltonian form. This transformation is beneficial for comprehending the fundamental characteristics of the system, but also serves as a powerful mathematical tool for further control and optimization endeavors.
(6)$$\dot{X}=f\left(X\right)$$

In the realm of multibody dynamics modeling, the process is often complex. Specifically, for tendon-driven, multi-rod-driven, and magnetic drive continuum and magnetic soft robots, it becomes necessary to incorporate the descriptions of tendons, rods, or magnetic fields, and establish their relationships with the generalized coordinates. Furthermore, additional elements may need to be considered to develop a more comprehensive dynamical model. For instance, tendons [106], multi-rod [107] and magnetic [108] elements. Sometimes, introducing Lagrangian multipliers, as suggested in [109], is required to accurately describe these interactions in the model. An interdisciplinary and multifaceted approach is often necessary for more complex scenarios, considering various factors such as environmental constraints, as detailed in [110]. It is important to note that even with a comprehensive model, there are inherent assumptions and limitations. For instance, some models might assume material homogeneity or overlook nonlinear factors like friction and air resistance. Therefore, understanding the assumptions and limitations of these models is crucial when applying them in practical scenarios.

### 2.2. Data Modeling

Traditional rigid robots have been primarily utilized in factory settings, focusing on executing single, predefined tasks. Precise mathematical models are often one of the best options for these applications. However, as the tasks and environments for robotic applications become more complex, researchers have attempted to develop more intricate models. Yet, this approach significantly increases computational costs. In practical applications, compromises often need to be made, followed by optimization through control algorithms, which may not fully leverage the potential of modeling techniques. The challenge of modeling and controlling compliant continuum robots designed to operate in complex environments is substantial. Initially, the focus was primarily on developing models based on various assumptions.

With the ascent of deep learning [111,112,113] and artificial intelligence [114], data-driven models have garnered widespread attention across multiple domains, including robotics [115,116,117]. These models are increasingly being integrated into robotic modeling processes. Soft robots have notably adopted these advanced technologies, achieving significant breakthroughs [56,118,119]. This trend has also captivated researchers in continuum robots, a field grappling with nonlinear modeling challenges, spurring extensive research into data-driven modeling methodologies for continuum robots [48,120,121]. Data-driven modeling relies heavily on collecting and preprocessing high-quality data and selecting features and models carefully. In the context of continuum robotics, data acquisition predominantly depends on sensor data [122,123] (such as position, shape, flexibility, and bending), control signals, external databases or systems [124] (like SOFA [125], Sorosim [126], and SimSOFT [127]), nonlinear experimental data [128], simulation data [129,130,131], particular environmental factors, and expert input.

In data-driven modeling, particularly in the application to continuum robots, subsequent steps and corresponding challenges arise once data collection is completed. These steps include data preprocessing [132,133], feature engineering [134,135], model selection [136,137], model training [138], model validation [139] and, ultimately, model deployment [140]. For instance, challenges such as addressing missing and outlier values often arise during the data preprocessing stage, which is typically complex and prone to errors. Feature selection and engineering require an in-depth analysis of the raw data to identify the most relevant features. Meanwhile, during the model selection and training phases, we encounter the intricate task of choosing the most suitable model for the problem and fine-tuning its parameters.

Research and practice have adopted various effective strategies to address complex issues. During the data preprocessing stage, statistical methods and professional cleaning tools are employed [141]. Machine learning assesses feature importance and conducts correlation and causality analyses for feature selection. Model selection and training heavily rely on cross-validation and grid search techniques. Regularization or ensemble methods are utilized during the model validation phase to prevent overfitting. Finally, model deployment involves A/B testing to verify real-world utility and performance monitoring to ensure stability. Data-driven modeling, especially in applying continuum robots, confronts various challenges. These include, but are not limited to, data quality, high dimensionality and sparsity, imbalanced datasets, and the optimization of model hyperparameters. Furthermore, computational resource limitations and model interpretability must also be considered. Specific techniques and approaches must be employed to ensure the effectiveness and reliability of the models.

Various machine-learning models have been successfully employed in various application scenarios of continuum robots. These models include neural networks [142,143,144], reinforcement learning [145], support vector machines [146], and a myriad of combined strategies [147]. They have demonstrated exceptional performance in trajectory prediction, action recognition, and fault detection. Moreover, statistical models like Bayesian networks and Gaussian processes have also played a role in estimating the state and parameters of robots.

However, it is noteworthy that in the application of continuum robots, the interpretability of models [148,149,150] holds importance. This is especially evident in critical application scenarios such as medical surgery, where understanding the logic behind model predictions enhances user trust in the model and is also a critical factor in ensuring operational safety. Yet, deep learning models are often perceived as ’black boxes’ with complex internal logic to decipher. This challenge extends beyond technical aspects, encompassing ethical, social, and legal dimensions, suggesting that a comprehensive solution may involve a broader range of disciplines.

An interdisciplinary perspective, particularly from fields such as computer science, ethics in artificial intelligence, and psychology, offers new directions and methodologies for addressing the issue of model interpretability [151,152,153]. Integrating concepts like attention mechanisms [154] and local interpretable models can uncover the rationale behind model decisions [155]. This not only enhances the credibility of models in applications such as continuum robots, but also takes into account the ethical and social responsibilities of the models. In applying continuum robots, data-driven modeling is pivotal in solving technical challenges and opens new avenues for interdisciplinary research and collaboration. This contributes not only to the expansion of application horizons, but also provides new perspectives and tools at both theoretical and practical levels for addressing complex problems in the real world.

### 2.3. Hybrid Modeling

Principle modeling typically focuses on deriving fundamental equations of robot kinematics from basic physical principles. Still, such models often necessitate simplifications or assumptions in dealing with complex factors, such as friction and nonlinear responses. Conversely, data-driven modeling relies on extensive information collected from experimental data or real-world operations, fitting or interpreting these data through machine learning or statistical methods. Yet, it may lack a profound understanding of the underlying physical processes. Hybrid modeling [156,157] aims to synthesize the strengths of both approaches, thereby achieving a more comprehensive and accurate representation of intelligent system behavior.

Hybrid modeling represents a multi-scientific amalgamated modeling strategy [158], integrating diverse modeling methodologies and data sources [159,160]. This includes, but is not limited to, physically based models, data-driven models, statistical models, heuristic algorithms, and expert knowledge. The strategy aims to achieve comprehensive and precise description and control of complex, uncertain, and nonlinear systems by amalgamating various sources of information. The framework is applicable in the narrow sense of combining physical and data models and in a broader context of blending interdisciplinary modeling approaches [161]. Hybrid modeling in continuum robots primarily focuses on incorporating data-driven elements into physical models, particularly in the aspect of control algorithms [162]. Although the efficacy of this hybrid method has been notably enhanced with the continuous advancement of principle models and data science technologies [163], the significant compliant nonlinearity characteristics of continuum robots and the complexity of their operating environments necessitate and urge the expansion of the application scope and perspective of hybrid modeling.

Hybrid modeling has been extensively researched across various disciplines [157,164,165,166,167]. For the first time, we explore the hybrid modeling of continuum robots from both vertical and horizontal perspectives. A key element in the vertical approach is determining how to allocate weights to theoretical and data models appropriately, a process often dynamic and dependent on the environment. In scenarios with insufficient experimental data or low data quality, theoretical modeling is usually given greater weight, leveraging existing physical knowledge and mathematical theories for more reliable predictions. Conversely, when data are abundant and reliable, data models may receive higher weighting to capture complex environments’ impacts or nonlinear factors’ impacts more accurately. Additionally, in the framework of hybrid modeling, the horizontal integration strategy is also crucial, involving the combination of different types or sources of models on the same level [168,169,170,171]. For example, a continuum robot may possess multiple degrees of motion and sensory modules, each capable of being modeled theoretically and through data independently. Horizontal integration then addresses how to amalgamate these independent or partially overlapping models into a unified, more comprehensive model.

The hybrid modeling approach may increase the complexity and computational cost of the model while also complicating the model validation process. Ensuring that theoretical and data models are based on consistent assumptions and datasets to maintain data consistency presents a challenge [165,172]. Dynamically adjusting model weights can enhance adaptability, but may also impact model performance. Additionally, in an interdisciplinary environment, model interpretability should not be overlooked [173]. Resolving potential disciplinary contradictions or conflicts is a complex yet necessary task. Hybrid modeling provides a possible theoretical framework for continuum robots and extends to a more interdisciplinary domain. Within the broader context of interdisciplinary research, hybrid modeling could emerge as a diversified framework, accommodating knowledge and methodologies from various fields ranging from physics and material science to computer science, robotics, and statistics. This not only accelerates the flow of information and exchange of knowledge between disciplines, but also enriches the interdimensionality and accuracy of the models. More importantly, such interdisciplinary collaboration implies a multi-faceted examination of model assumptions and limitations, enhancing the model’s reliability and adaptability.

## 3. Magnetic Soft Robots

While continuum robots focus on millimeter-scale or more oversized dimensions, magnetic soft robots can extend to the nanoscale. However, ignoring the quantum effects of microscopic physical phenomena becomes challenging at the nanoscale. Therefore, the influences of different forms of magnetic fields and quantum effects are equally important to consider.

### 3.1. Uniform Magnetic Field

The uniform magnetic field is essential for its stable control environment in magnetic soft robots. This stability simplifies experimental design and ensures predictability and repeatability in wide-ranging applications, highlighting the need for advanced modeling to leverage its unique benefits effectively. For the magnetic soft robots described in Equation (Equation 5), the primary source of actuation has shifted from mechanical drive to the torque exerted by magnetic moments. This transition simplifies the model and opens new possibilities for precise control. Specifically, based on the existing continuum robot dynamics models, we can construct a more comprehensive and unified theoretical framework for magnetic soft robots in uniform magnetic fields by introducing magnetic moments as the main source of actuation [108,174,175]. For instance, the interaction between the magnetic moment m and a uniform magnetic field B can be described by the following mathematical expression involving magnetic field strength, current density, and other physical parameters:(7)$$\tau_{\mathrm{mag}}=f\left(m,B\right)=m\times B$$

The magnetic moment term in Equation (Equation 7) needs to be incorporated into Equation (Equation 5) to successfully construct the dynamic model of filamentous magnetic soft robots. This model increases the complexity and comprehensiveness of the original dynamics model, and opens new possibilities for precise control and optimization. Further information on the construction of filamentous magnetic soft robots can be found in the related literature [176,177,178,179].

### 3.2. Non-Uniform Magnetic Field

Despite the preference for uniform magnetic fields due to their simplicity in modeling and predictability in operational contexts, such as in the case of filamentous magnetic soft robots [25], non-uniform magnetic fields have demonstrated undeniable advantages in specific specialized medical applications. Specifically, non-uniform magnetic fields offer enhanced capabilities for localized and adaptive manipulation, making them particularly suitable for interventions in complex and deep-seated tissue structures, such as aortic treatment (Figure 1-3a) [180], cancer therapy [181,182,183], neuro intervention (Figure 1-3b) [184], intravascular surgery (Figure 1-3c) [185,186], and endoscopic procedures (Figure 1-3d) [187], etc. [188]. These unique advantages underscore the critical importance of non-uniform magnetic field modeling in medical scenarios requiring high precision and flexibility in deploying soft magnetic robots.

In a uniform magnetic field, since the net magnetic force is zero, our discussion primarily focuses on the influence of the magnetic torque. However, when transitioning to a non-uniform magnetic field, the situation becomes more complex. In such environments, microrobots are influenced not only by magnetic torque but also by magnetic forces. This can be expressed by the following equation, which demonstrates that:(8)$$F_{\mathrm{mag}}=\nabla \left(m\cdot B\right)$$

Although the hybrid Equation (Equation 8) increases the complexity of the model, it also expands our capability to control magnetic soft robots in various application scenarios precisely. Furthermore, fluid resistance becomes an indispensable dynamic factor in scenarios involving fluid mediums, such as operations within blood vessels or body cavities, especially in applications involving the manipulation of microrobots in fluid mediums. The following equation can represent this resistance:(9)$$F_{\mathrm{fluid}}=-6\pi \eta r\left(v-u\right)$$

With a viscosity of η, u is the fluid velocity and v is the velocity of the robot in the fluid, and *r* is the approximate radius of the robot. Considering fluid resistance makes the multiphysics model more aligned with real-world applications and provides rich content for subsequent in-depth analysis and understanding. The net external force generated by the magnetic field and fluid resistance is reflected in the acceleration $\frac{d^{2}x}{dt^{2}}$ of the robot’s center of mass. The latter describes the robot’s angular acceleration $\frac{d^{2}\theta }{dt^{2}}$ around its center of mass, which is determined by the total external torque τ applied. These two equations provide us with a complete and in-depth perspective for understanding and analyzing the dynamic behavior of robots in complex multiphysics fields. Therefore, the motion equation and rotational dynamics of the robot are, respectively, given by:(10)$$m\frac{d^{2}x}{dt^{2}}=F_{\mathrm{mag}}+F_{\mathrm{fluid}}\;I\frac{d^{2}\theta }{dt^{2}}=\tau$$

It should be noted that fluid resistance, the mass matrix, and the Coriolis terms remain constant in both models. However, we often face more complex magnetic field environments in practical applications. These environments may not only be non-uniform, but may also involve the combined effects of multiple magnetic fields. More importantly, in actual surgical applications, it is necessary to consider problems faced by interdisciplinary, such as biofilms [189] and infections related to catheters [190,191]. Although the literature [192] proposes a strategy for preventing biological infections, it still confronts multiple challenges [193]. Therefore, realizing the application of magnetic soft robots in the medical field requires interdisciplinary collaboration and integration.

### 3.3. Quantum Effects

At the micro and nano scales, modeling magnetic soft robots particularly requires further consideration of aspects such as quantum effects and molecular dynamics, as these factors may play a significant role at this scale [194]. For instance, quantum effects could influence the electromagnetic properties of materials [195,196]. Therefore, it is necessary to select a quantum mechanical model to describe these phenomena in addition to the dynamic description provided by Equation (Equation 10). This could include models like Density Functional Theory [197,198] (DFT) or Hartree–Fock [199,200], among others. This model is typically defined by a Hamiltonian $H_{Q}$:(11)$$H_{Q}=T+V_{Q}\left(r_{Q}\right)$$
where *T* represents the kinetic energy term and $V_{Q}\left(r_{Q}\right)$ is the quantum potential energy. The system’s ground state or several low-excited states are found by solving the Schrödinger equation or other quantum equations corresponding to the Hamiltonian $H_{Q}$. Subsequently, the quantum correction force $F^{Q}$ is calculated, which is typically the gradient of the quantum potential energy $V_{Q}$ concerning the coordinates $r_{Q}$:(12)$$F^{Q}=-\nabla V_{Q}\left(r_{Q}\right)$$

This approach of introducing quantum effects through quantum correction forces offers the advantages of simplicity and broad applicability. Still, it also has the drawbacks of limited accuracy and the potential for increased computational burden. Finally, it is worth noting that in addition to quantum correction forces, path integral molecular dynamics (PIMD) can be used for a careful consideration of quantum effects [201,202]. PIMD represents a more exhaustive yet complex method, typically employed in systems where precise consideration of quantum effects is necessary.

The complex response characteristics of magnetic soft robots in nonlinear magnetic fields increase the difficulty of data modeling, rendering traditional linear models inadequate. Nonlinear models or deep learning algorithms are necessary to capture these relationships [203]. Modeling of magnetic soft robots must address time dependency, potentially utilizing networks with memory capabilities such as RNNs or LSTMs. Three-dimensional operations and complex magnetic fields pose challenges for data collection, necessitating specialized sensors or computer vision techniques. Data modeling [204,205,206] and hybrid modeling offers multiple options for magnetic soft robots, in contrast to the mature technologies of continuum robots. Researchers should draw on continuum robot strategies, emphasizing the integration of precise models, advanced algorithms, and sensing technologies while focusing on interdisciplinary biocompatibility studies in biological environments.

Data modeling for magnetic soft robots poses more significant challenges than traditional continuum robots, necessitating the management of more complex issues such as data sparsity imbalance and ensuring model interpretability and safety. Models must accurately capture nonlinear magnetic responses and maintain reliability in dynamic environments. This requires integrating data science and physics knowledge, advanced deep learning, and physical models to ensure accuracy in their three-dimensional operations and complex magnetic field responses. Therefore, interdisciplinary hybridization and combining theoretical and practical data are crucial in developing magnetic soft robots.

### 3.4. Numerical Framework

Following a detailed exploration of the interdisciplinary modeling framework for continuum robots and magnetic soft robots, numerical simulation emerges as a critical step in realizing these models. Discretization is often necessary to enhance the programmability of the robot models [207]. To meet the complex demands of interdisciplinary research, we have meticulously developed a novel classification strategy based on a theoretical perspective of basis functions. This categorization divides numerical methods into three major types (Figure 2): basis function methods, zero basis function methods, and hybrid zero basis methods. Within basis function methods, we further distinguish between global basis functions (such as spectral methods), local basis functions (like finite element methods), and hybrid methods (e.g., spectral-element methods). Zero basis function methods primarily encompass a range of specific algorithms, including Boltzmann lattice and Monte Carlo methods. Meanwhile, hybrid zero basis methods include innovative approaches to multiscale or interdisciplinary issues, particularly suited for complex problems such as fluid–structure interaction.

Although a plethora of literature has provided non-specialist readers with theoretical overviews of continuum mechanics [41,124,208,209,210] and magnetic soft robotics [211], offering novices in the field a broad perspective, interdisciplinary researchers still face challenges in selecting appropriate numerical methods and implementing them for numerical solutions. In this context, commercial simulation platforms such as Abaqus [212] and COMSOL Multiphysics [213], with their user-friendly interfaces and extensive case libraries, have emerged as powerful tools in interdisciplinary research, significantly lowering the barriers to entry. However, while these platforms have streamlined the numerical simulation process, a thorough understanding of the underlying mathematical principles remains crucial for expanding the frontiers of interdisciplinary integrated research. By deepening their knowledge of the mathematical framework, researchers can address complex problems more innovatively and foster the amalgamation of interdisciplinary expertise.

## 4. Interdisciplinary Analysis

### 4.1. Integration Analysis

Mathematical models are pivotal across multiple disciplines, including biomedical engineering, material science, chemistry, computer science, and pharmacology. In biomedical engineering, for instance, magnetic soft robotics models are instrumental in predicting interactions with complex biological tissues, offering precise simulations of cellular growth dynamics crucial for tissue engineering [214,215]. Within material science, these models aid in forecasting the performance of novel magnetic materials, particularly under extreme conditions [216]. In chemistry, models accurately delineate drug molecules’ propagation and reaction kinetics in complex systems, providing vital information for drug design [217,218]. In computer science, optimized algorithms utilize mathematical models to enhance the maritime capabilities of robots in unknown environments [219]. Lastly, in pharmacology, mathematical models are crucial for the design of personalized medication treatment plans, guiding dosage selection, and the development of treatment strategies [220].

Despite the extensive applicability of mathematical models across various disciplines, they exhibit notable limitations [221,222]. In biological applications, models often fail to capture the full complexity of biological systems, such as nonlinear interactions among multiple cells [223]. In material science, models may not adequately account for defects and impurities in materials during the manufacturing process [224]. Models of chemical reactions have limitations in predicting multiple reaction pathways, especially under variable experimental conditions [225,226]. In computer science, navigational algorithms may not be sufficiently adaptable to the variable and uncertain factors encountered in real-world environments [219]. In pharmacology, models also demonstrate limitations in considering individual genetic differences in drug responses [227]. Therefore, while these models provide valuable theoretical frameworks, they require continual refinement and validation by integrating experimental data and interdisciplinary knowledge.

In addressing the limitations of models, different disciplinary fields have developed their unique resolution strategies. Biologists utilize systems biology and high-resolution imaging techniques to incorporate detailed cellular and molecular level data into models, capturing the dynamics of complex biological systems [228,229]. Material scientists refine models by integrating multiscale simulations and high-throughput experimental data [230], detailing models to reflect micro defects and macroscopic properties during material fabrication [231]. Experts in the field of chemistry employ quantum chemical computations [232,233] and chemical kinetics simulations [234] for more precise predictions of reaction pathways and model calibration through experimental data. In electronics engineering and computer science, machine learning and data-driven approaches are used to enhance the adaptability of algorithms to cope with uncertainties in complex environments [71]. Meanwhile, pharmaceutical research is turning towards personalized medicine, integrating genomic information [235,236] and patient-specific biomarkers [237] to tailor models for accurate prediction of drug efficacy. The common goal of these strategies is to enhance the generalizability of models, ensuring that theoretical predictions better serve practical applications while promoting deeper interdisciplinary collaboration.

Faced with the limitations of models in specialized disciplines and the complexities of real-world application environments, in-depth research within a single discipline, despite its technical sophistication, often struggles to meet the challenges of practical applications fully [238,239,240,241]. The complexity of real-world applications demands models that are theoretically precise and possess interdisciplinary adaptability and applicability. In such contexts, interdisciplinary, integrated research becomes necessary for solving complex problems. More comprehensive models can be developed by integrating expertise in biology, material science, chemistry, computer science, and pharmacology. These models maintain their effectiveness and flexibility in the face of the variability and uncertainties of real-world applications. Interdisciplinary collaboration contributes to the empirical validation and improvement of models and the advancement of innovative technologies, ensuring the smooth translation of research findings into practical applications. Therefore, building an interdisciplinary collaborative platform to facilitate knowledge sharing and technology has become necessary in scientific research and technological innovation [242].

In interdisciplinary research, combining specialized technology with mathematical models is vital to enhancing precision and efficiency. In biology, high-throughput sequencing offers a wealth of genetic data, bolstering the accuracy of gene expression predictions [243]. Material science employs nanotechnology, such as atomic force microscopy, to refine models for accurately reflecting microscopic physical properties [244,245]. In chemistry, real-time monitoring techniques like mass spectrometry [246] provide direct data for kinetic models, optimizing reaction predictions. Deep learning algorithms in computer science process large datasets to reveal data patterns, guiding model adjustments [247]. Meanwhile, in pharmacology, combining clinical data with pharmacokinetic models supports the formulation of personalized treatment plans [248]. This melding of technology and models deepens disciplinary understanding and plays a significant role in technological advancement and application translation.

### 4.2. Case Analysis

To further study the interdisciplinary integration of continuum and magnetic soft robots, this article selects the research of Professor Zhao’s team as a case study. Moreover, numerous distinguished groups, such as those cited in [4,249,250,251,252,253,254,255,256], have demonstrated exceptional interdisciplinary integration capabilities in the research of continuum and magnetic soft robots, contributing to significant advancements within the field.

Initially confronting the emerging field of magnetic soft robots, Professor Zhao made pioneering contributions in the early stages, laying an essential foundation for the development of the field. In exploring novel soft materials, their work focused on hydrogels (Figure 3a) [257,258,259,260,261,262,263] and dielectric materials [264], addressing numerous challenges in theoretical modeling [265,266,267] and functionalization [268]. By combining these advanced materials with innovative manufacturing technologies (Figure 3b) [258,269,270], Professor Zhao and his team’s research outcomes have established a solid foundation for both the theoretical development of magnetic soft robotics and the manufacturing techniques of advanced materials. Their subsequent breakthroughs in magnetic soft robotics provide a robust accumulation of scientific and technological advancements.

Since 2018, the research team, building on their extensive experience in foundational theories [267,271], soft materials [272,273], and advanced manufacturing technologies [269], embarked on a systematic study of magnetic soft robotics. Utilizing the combination of 3D printing technology and magnetic media, they achieved innovations not only in the fabrication of magnetic soft robots (Figure 3c) [66], but also made significant contributions to the foundational theory and methodologies in magnetomechanics, providing new theoretical frameworks and computational methods for magnetoelastic mechanics (Figure 3d) [88,176,274,275]. Subsequently, the team effectively integrated material science, mechanical engineering, and computer science knowledge to develop magnetic soft robots with innovative features (Figure 3e) [25]. Following these initial achievements, Professor Zhao’s team also explored the application of magnetic soft robots in the biomedical field, particularly in neurovascular interventional treatments (Figure 3f) [69], demonstrating their broad applicability in interdisciplinary applications. This series of research efforts reflects the team’s in-depth exploration and practice in integrating interdisciplinary applications.

**Figure 3 micromachines-15-00313-f003:**
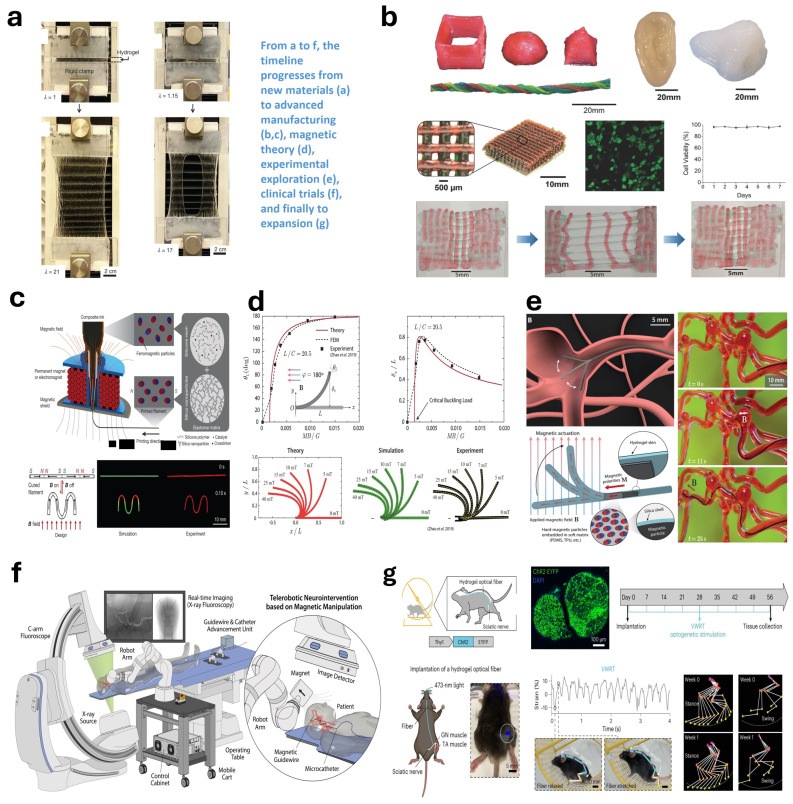
Research cases. In this interdisciplinary research case, the team initially focused on the enhancement of hydrogel properties. (**a**) Aiming to improve its physical characteristics, such as tensile strength. Subsequently, utilizing advanced manufacturing technologies, the team adeptly transformed the improved hydrogels into complex structures [257]. (**b**) This not only validated the practical utility of the material, but also propelled the development of manufacturing techniques. Further attempts were made to 3D print programmable ferromagnetic domains in soft materials [258]. (**c**) Yielding substantial academic achievements as illustrated. Following this, the team delved into the study of magnetorheological theory in flexible materials [66]. (**d**) Providing crucial scientific underpinnings for the design of magnetorheological soft robots. Building on these theoretical and material advancements, the team constructed and tested a prototype of the magnetorheological soft robot [176]. (**e**) Demonstrating an effective integration of theory and practice. Additionally, they extended the application of the magnetorheological soft robot to clinical experiments in the medical field [25]. (**f**) Exploring its potential in medical applications. In a lateral expansion of their research, the team also developed hydrogel fibers with high fatigue strength [69]. (**g**) A technology that holds broad prospects in optogenetics [276].

In his extensive research across multiple disciplines, Professor Zhao, taking the study of magnetic soft robots as an example, has integrated forefront technologies and knowledge from material science, mechanical engineering, physical chemistry, and biomedicine, showcasing the depth and breadth of his research. In terms of interdisciplinary integration, his team has advanced innovations in the use of 3D printing technology, not only in the development of hydrogel fibers (Figure 3g) [276] and conductive polymers [277], but also in pioneering explorations in the field of bioelectronics, such as the development of 3D printable high-performance conductive polymers for all-hydrogel bioelectronic interfaces [262,278]. Furthermore, Professor Zhao’s interdisciplinary research extends to the development of biological adhesives [279], significant for the sutureless repair of gastrointestinal defects [280]. His team has also achieved innovations in medical robotics, for instance, developing soft neural prosthetics [281] that offer electromyography control and tactile feedback, significantly enhancing the naturalness and user experience of prosthetic technology. In biomedical imaging, Professor Zhao’s team’s bioadhesive ultrasonic technology offers new solutions for long-term continuous imaging of various organs [282], holding significant potential for disease monitoring and surgical navigation.

## 5. Discussion

This article explores the challenges continuum, and magnetic soft robots face in medical applications, analyzing them from the perspective of model construction to interdisciplinary, integrated applications. We combine the knowledge of topology and group theory to build a unified model framework covering everything from continuum robots to magnetic soft robots. This framework promotes interdisciplinary learning and communication and provides a basis for in-depth discussion of different robot design and application disciplines’ issues, impacts, and limitations. Furthermore, through case analysis, this article reveals the importance of moving from basic theory (including model construction) to interdisciplinary comprehensive application in addressing the challenges of continuum and magnetic soft robots in the medical field.

Current developments in continuum and magnetic soft robotics exhibit two notable trends. On the one hand, many researchers are actively leveraging the latest outcomes of cutting-edge technologies [283], focusing on finding solutions within their specific disciplines [79,213,284,285]. However, this approach often overlooks critical interdisciplinary factors. For instance, in studies concerning the use of robots in blood environments, many have not adequately considered how environmental factors affect the functionality and safety of the robots. On the other hand, while some studies attempt to blend knowledge from multiple disciplines, including magnetomechanics, advanced manufacturing technologies [286,287,288], material science, and chemistry, there are still unresolved issues regarding essential materials. These include the biocompatibility of neodymium-iron-boron in applications [25] and clinical efficacy issues like biofilm infections in hydrogel thin films. Although these issues have garnered the attention of biomedical researchers [192], challenges remain regarding such technologies’ mechanisms and effective control.

Faced with these challenges, it is essential to recognize that while single-discipline research may be efficient in certain situations, interdisciplinary collaboration becomes particularly crucial in practical applications. Such cooperation facilitates the exchange and integration of knowledge across different fields and effectively addresses issues that might be overlooked from a single-disciplinary perspective. For instance, in the selection of materials, by combining expertise from material science, biomedicine, and mechanics, we can more comprehensively assess the suitability and safety of materials. Simultaneously, interdisciplinary teams can collaboratively explore new design solutions, such as developing novel composite materials, to meet the demands of robotic technology in complex environments.

However, despite the foundation of continuum and magnetic soft robotic design being rooted in advanced materials [278,289,290] and technologies [291,292], significant challenges arise in practical applications, particularly during sensitive operations such as intricate surgical procedures. These challenges are primarily manifested in the areas of structural design and control application. For instance, magnetic soft robots provide critical insights into the miniaturization of continuum robots, yet both remain in the nascent stages of academic research, facing heightened demands in real-world applications. The prolonged review process for medical devices intended for human intervention undeniably poses an obstacle, yet it does not constitute the crux of the issue.

The central issue lies in the fact that current research efforts are predominantly confined to individual disciplines or limited interdisciplinary studies. In modeling, researchers might focus on enhancing the accuracy of models, leading to complexities that render them unsuitable for real-time control. Therefore, it often becomes necessary to simplify these models and compensate for control precision through sensor feedback. Simultaneously, in sensor research, despite a focus on precision and stability, the biocompatibility of sensors in the complex, unstructured human body environment is often overlooked. Even studies that consider biocompatibility fail to fully address issues like biofilm infections during catheter-based interventions. Additionally, considerations around privacy, technological iteration, commercialization, and legal challenges must be taken into account [293,294]. Hence, to transition continuum and magnetic soft robots from academic research to technological application, deep, interdisciplinary collaboration becomes crucial. This inherently demands that each research phase provide effective ’interfaces’ (or meta-questions), facilitating in-depth synergy and knowledge exchange among various disciplines, thereby enabling a more comprehensive and efficient scientific inquiry.

## 6. Conclusions

This review provides a comprehensive review of the challenges of continuum and magnetic soft robotics in medical applications, particularly emphasizing the importance of interdisciplinary approaches in developing this field. Through a comprehensive analysis, we demonstrate the critical role of algebra and geometry in building a unified model framework. At the same time, data modeling and hybrid modeling are discussed, and their implications for precise control and practical applications are pointed out. Furthermore, this review reveals the potential for comprehensive interdisciplinary research to improve the utility and effectiveness of medical robots. Therefore, further strengthening interdisciplinary research and cooperation will be key to promoting technological innovation in this field and solving practical application challenges.

## Figures and Tables

**Figure 2 micromachines-15-00313-f002:**
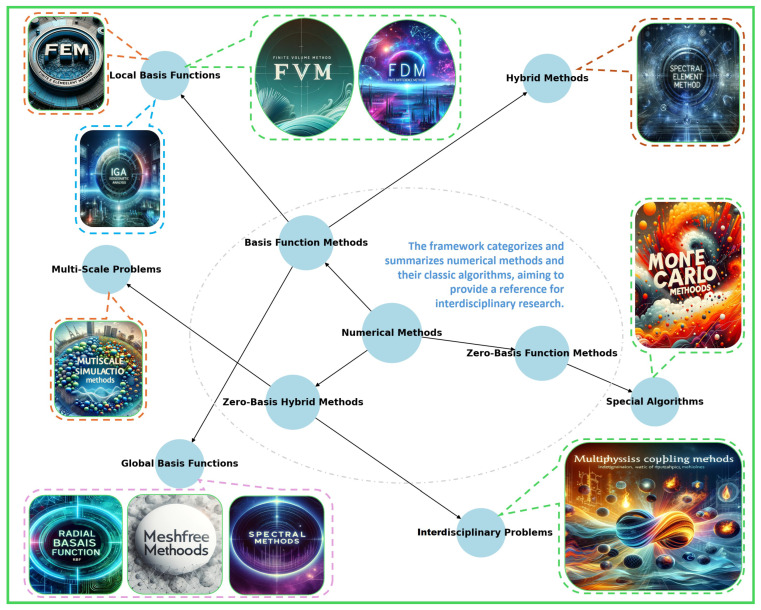
Numerical methods perspective. Numerical methods are pivotal in transforming theoretical models into executable computational paradigms. This process is paramount in interdisciplinary domains such as robotics, which involves converting abstract theoretical concepts into practical computational procedures. From the perspective of basis functions, we categorize numerical methods into three primary classifications: basis function methods, zero basis function methods, and hybrid zero basis methods. This categorization not only aids in identifying and comprehending the characteristics and applicable contexts of various numerical techniques but also highlights their central role in interdisciplinary, integrated analysis. For instance, in robotics, these methods facilitate more precise simulation and analysis of robotic dynamics, sensory systems, and environmental interactions. By delving into the role of these numerical methods, including developing disciplines, we enhance our understanding of these techniques and establish a more robust and efficient computational foundation for robotics and a broader spectrum of scientific disciplines.

## Data Availability

Data is contained within the article.

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
