# Peer review of "Continuum Robots and Magnetic Soft Robots: From Models to Interdisciplinary Challenges for Medical Applications"

_micromachines, 2024, doi:10.3390/mi15030313_

Round 1
Reviewer 1 Report
Comments and Suggestions for Authors
The authors reviewed modeling covering continuum and magnetic soft robots. While the review addressed various modeling techniques, it lacks a logical integration of interdisciplinary challenges within the discussion. The authors need to specifically organize how modeling approaches and interdisciplinary challenges affect continuum and magnetic soft robots.
Q) Authors discussed magnetic soft robots with a focus on continuum and flexible soft robots. Considering that continuum soft robots are mainly fabricated on the centimeter-to-meter scale for medical applications, it raises questions about the relationship between the main discussion and quantum effects. It is difficult to understand the necessity of considering quantum effects for these larger-scale continuum soft robots.
Q) Authors addressed the single case analysis from Zhao group. However, for a comprehensive review of continuum soft robots, it is crucial to discuss relevant developments from other groups in the field, organize the research trends, and provide perspectives.
Q) Figure 2 did not explain the respective numerical method although they were visualized.
Q) To enhance reader comprehension, the authors need specific explanations for certain statements. For example, in line 43, the term 'rapid application of continuum robots' is mentioned. Could you please provide clarification in this context? Does 'rapid' refer to fast growth in continuum robotics or quick movements of continuum robots that have adaptability or perhaps another aspect?
Q) The schematic in Figure 1-2 shows the chemical morphology of polymer composites rather than the structure of medical MSRs. For consistency, I suggest the revision of the figure.
Comments on the Quality of English LanguageQ) Please double-check the manuscript for grammatical errors to increase readability. In lines 67 and 79, the capitalization and use of small letters need to be reviewed.
Author Response
Dear reviewer:
On behalf of my team, I would like to express my sincere gratitude to you. Your review of our paper was rigorous and detailed and provided valuable suggestions and feedback. We have thoughtfully discussed the issues and suggestions you pointed out, and have carefully responded and corrected each opinion. The specific responses and modification details are in the attachment for your reference.
We thank you again for your professional guidance and careful review. Regarding the importance and quality of this research, we look forward to continuing to receive your valuable guidance and feedback to improve our work further.
Best regards
Honghong

Reviewer 2 Report
Comments and Suggestions for Authors
The review paper delves into the challenges posed by continuum and magnetic soft robotics in the realm of medical applications, offering intriguing and meaningful insights from various perspectives. Some suggestions are listed below:
11. Given that surgeons and physicians may also find the paper relevant, the authors may opt to provide a brief explanation of the working principles and fundamental concepts behind continuum robots and magnetic soft robots in the introduction. This approach would facilitate a better understanding of the content for individuals with backgrounds in medical or biological fields.
22. To enhance the presentation of the modeling efforts for continuum robots and magnetic robots, the authors could consider employing a structured format that chronologically lists the progress made in modeling. Additionally, this format could include an evaluation of the advantages and disadvantages of each modeling approach. Such a systematic approach would provide readers with a clear overview of the development timeline and a comprehensive understanding of the strengths and limitations of each modeling technique.
33. The development and research of human-based medical devices typically span multiple disciplines, promoting interdisciplinary learning and communication to facilitate in-depth discussions about design. In the discussion, it is crucial to provide a comprehensive summary that highlights the unique challenges and advantages faced by continuum and magnetic soft robots compared to those encountered by other medical devices, such as commercially available medical robots, from various perspectives.
Comments on the Quality of English LanguageThe English language used in the paper is of standard quality.
Author Response

(The authors gave the same response as above.)

Round 2
Reviewer 1 Report
Comments and Suggestions for Authors
Figure 3: To improve overall readability and clearly convey the aimed information that was analyzed to be the research case, the authors can consider resizing the respective images or adjusting the text size.
Comments on the Quality of English Language
Author Response
Dear reviewer:
Thank you again for your valuable suggestions on our revised paper. In response to your questions and suggestions, we have made corresponding modifications, and the specific content has been marked in the attachment. We sincerely thank you for your valuable time and effort in reviewing our paper.
Best regards
Honghong

Reviewer 2 Report
Comments and Suggestions for Authors
In the third response, the author could provide a more detailed comparison between continuum/magnetic soft robots and commercially available medical robots. For instance, they could elucidate the specific reasons why soft robots are necessary and why they are designed in a continuum/magnetic fashion. While medical robots are required to be precisely controlled for surgical procedures, necessitating complex control algorithms, is it a similar level of intricacy is demanded for soft continuum/magnetic robots due to their non-linear deformation characteristics?
To enhance the comparison, the author should delve into the inherent challenges associated with the unique design and functionality of soft robots, rather than simply reiterating commonly known challenges faced by medical robots. By offering a comprehensive analysis, the author can provide a more insightful perspective on the distinct requirements and complexities of both types of robots.
Comments on the Quality of English LanguageThe English language is understandable.
Author Response

(The authors gave the same response as above.)
